# An evaluation of factors that may influence clinicians' decisions not to enroll eligible patients into randomized trials in critical care

**Mahesh Ramanan** [1,2,3,4] *, **Laurent Billot** [2], **Dorrilyn Rajbhandari** [2], **John Myburgh** [2], **Balasubramanian Venkatesh** [2,4,5,6]

**1** Intensive Care Unit, Caboolture Hospital, Brisbane, Queensland, Australia, **2** Critical Care Division, The George Institute for Global Health, University of New South Wales, Sydney, New South Wales, Australia, **3** Adult Intensive Care Services, The Prince Charles Hospital, Brisbane, Queensland, Australia, **4** School of Medicine, University of Queensland, St Lucia, Queensland, Australia, **5** Intensive Care Unit, Wesley Hospital, Auchenflower, Queensland, Australia, **6** Intensive Care Unit, Princess Alexandra Hospital, Woolloongabba, Queensland, Australia

* Mahesh.ramanan@health.qld.gov.au

**Data Availability Statement:** Data can be made available on request pending approvals from the relevant Institutional Review Board, the Management Committee of the Australia New

## Abstract

### Objectives

To determine the association between intensive care unit (ICU) characteristics and clinicians' decision to decline eligible patients for randomization into a multicentered pragmatic comparative-effectiveness controlled trial.

### Methods

Screening logs from the Adjunctive Glucocorticoid Therapy in Septic Shock Trial (ADRENAL) and site-level data from the College of Intensive Care Medicine and Australia New Zealand Intensive Care Society were examined. The effects of ICU characteristics such as tertiary academic status, research coordinator availability, number of admissions, and ICU affiliations on clinicians declining to randomize eligible patients were calculated using mixed effects logistic regression modelling.

### Results

There were 21,818 patients screened for inclusion in the ADRENAL trial at 69 sites across five countries, out of which 5,501 were eligible, 3,800 were randomized and 659 eligible patients were declined for randomization by the treating clinician. The proportion of eligible patients declined by clinicians at individual ICUs ranged from 0 to 41%. In the multivariable model, none of the ICU characteristics were significantly associated with higher clinician decline rate.

### Conclusions

Neither tertiary academic status, nor other site-level variables were significantly associated with increased rate of clinicians declining eligible patients.

Zealand Intensive Care Society Centre for Outcomes and Resource Evaluation (as outlined here: https://www.anzics.com.au/data-access-and-publication-policy/ ), and The George Institute for Global Health (as outlined here: https://www.georgeinstitute.org/data-sharing-policy )

**Funding:** The author(s) received no specific funding for this work.

**Competing interests:** The authors have declared that no competing interests exist.

## Background

Prompt enrollment of eligible patients into randomized controlled trials (RCTs) is recognized as a research priority [1]. Enrolment of patients into RCTs in critical care poses challenges including difficulties in obtaining consent because of their altered conscious state due to either underlying conditions or acute therapeutic interventions [2, 3]. Within this environment, attending clinicians may decline to enroll potentially eligible patients, which has been reported to occur for 3–15% of eligible patients [3, 4].

A detailed evaluation of the influence of site-level factors on the clinician decline rate, that is the proportion of eligible patients that are declined for randomization by the treating clinician, in critical care RCTs has not been performed. Some reasons provided by clinicians relate to the nature of the intervention and lack of equipoise. We hypothesized that factors such as the academic and tertiary status of the intensive care unit (ICU) and availability of dedicated research staff may be associated with reduced clinician decline rate.

We performed this study to determine associations between ICU characteristics and clinician decline rate from a large-scale, investigator-initiated, pragmatic, international RCT.

## Methods

Institutional Review Board approval was obtained from The Prince Charles Hospital Human Research Ethics Committee (LNR/2018/QPCH/44975).

Patient randomization data was extracted from the *Adjunctive glucocorticoid therapy in patients with septic shock* [5] (ADRENAL) screening log and linked to ICU characteristics obtained from the Critical Care Resources registry maintained by Australia New Zealand Intensive Care Society (ANZICS) Centre for Outcome and Resource Evaluation. For non-Australia and New Zealand sites, these data were extracted from the site selection and feasibility log. The screening log contained date and site at which each patient was screened, whether they were eligible, randomized, and reason for not being randomized. We defined tertiary ICU status as any ICU meeting criteria for a Level III ICU as per College of Intensive Care Medicine of Australia and New Zealand [6].

The primary outcome was a binary variable indicating whether an eligible patient was declined for randomization by the treating clinician and was analyzed using a multivariable mixed effects logistic regression model. The clinician decline rate was calculated as the proportion of eligible patients who were declined for enrolment by the treating clinician and was presented as median and interquartile range (IQR). Univariate analyses comparing characteristics of tertiary and non-tertiary sites were performed using Mann-Whitney U test and Fishers exact test for continuous and categorical variables. ICU-level variables (volume, i.e., number of admissions per annum; research coordinator full-time equivalent (FTE); after-hours research coordinator availability; dedicated director of research FTE; and ANZICS Clinical Trials Group membership) were entered as fixed effects and site as random effect, with patients nested within sites. Two models were created; one for Australian and New Zealand sites only (additional covariates were available for these sites) and one including all sites. The effect of the covariates on the risk of being declined for randomization was presented as an adjusted odds ratio (OR) with 95% confidence interval (CI) and p-value <0.05 considered significant.

## Results

The ADRENAL trial screened 21,818 patients from 69 sites, of whom 5,501 were eligible for randomization, and 3 800 were randomized. The characteristics of tertiary and non-tertiary ICUs participating in the trial are listed in Table 1. Of the 1,701 patients who were eligible but

**Table 1. Site-level characteristics of tertiary and non-tertiary ICUs that participated in the ADRENAL trial.**

|  | Non-tertiary (n = 35 ICUs) | Tertiary (n = 34 ICUs) | Total (n = 69 ICUs) | P † |
|---|---|---|---|---|
| Volume (admissions per annum), median (IQR) | 909.5 (633–1320) | 1737 (1244–2156) | 1244 (850–1762) | <**0.001** |
| Director of Research FTE*, median (IQR) | 0 (0–0.05) | 0 (0–0.23) | 0 (0–0.1) | 0.3 |
| Research coordinator FTE, median (IQR) | 1 (1–2) | 2 (1–3) | 2 (1–3) | **0.02** |
| After-hours research coordinator availability n (%) | 13 (37.1) | 13 (38.2) | 26 (37.7) | 0.9 |
| CTG membership* n (%) | 16 (55.2) | 23 (95.8) | 39 (73.6) | **0.001** |
| Country n (%) |  |  |  | 0.5 |
| Australia | 26 (74.3) | 19 (55.9) | 45 (65.2) |  |
| New Zealand | 3 (8.6) | 5 (14.7) | 8 (11.6) |  |
| Denmark | 0 (0) | 1 (2.9) | 1 (1.5) |  |
| Saudi Arabia | 1 (2.9) | 2 (5.9) | 3 (4.4) |  |
| United Kingdom | 5 (14.3) | 7 (20.6) | 12 (17.4) |  |
| CDR, median (IQR) | 0.09 (0–0.2) | 0.07 (0.02–0.19) | 0.09 (0.01–0.19) | 0.8 |

Definitions of abbreviations: ICU = intensive care unit; ADRENAL = Venkatesh et al. '*Adjunctive glucocorticoid therapy in patients with septic shock*' Trial;

IQR = interquartile range; FTE = full-time equivalent; CTG = Australia New Zealand Intensive Care Society Clinical Trials Group; CDR = clinician decline rate

\* Director of Research FTE and CTG membership data is for Australia and New Zealand sites only

\* † p values presented in this column have been generated from Fisher's exact test for categorical variables and Mann-Whitney U tests for continuous variables. Bold p-values are statistically significant

not randomized, 659 were declined for randomization by the treating clinician. Clinician decline rates ranged from 0 to 41%(median rate of 7%, IQR 2–19% for tertiary sites and 9%, IQR 0–20% for non-tertiary sites).

In the multivariable analysis (Table 2), tertiary status was not associated with a significantly reduced clinician decline rate (OR 0.89, 95% CI 0.35–2.31, p = 0.8 for all sites model; OR 0.52, 95% CI 0.17–1.59, p = 0.3 for Australia and New Zealand sites model). None of the other available covariates were significantly associated with clinician decline rate in either model.

## Discussion

The decision by a clinician to exclude a potentially eligible patient is a recognized cause of reduced enrolment into a clinical trial. Using a representative international, large-scale pragmatic RCT in critically ill patients, we report that nearly twelve percent of eligible patients were declined for enrolment by treating clinicians. This rate was not influenced by the academic status of the ICU nor other site characteristics. The ADRENAL trial was a pragmatic study evaluating a commonly used intervention and also allowed deferred consent (patients were allowed to be randomized into the trial and consent obtained subsequently).

The reasons for clinicians declining to enroll eligible patients were not collected in the screening log. The availability of adequate site level resources for trial conduct, the use of a pragmatic and user-friendly consent model and clinician familiarity with the intervention being tested suggest that these were unlikely contributors to the clinician decline rate.

The possibility that clinician equipoise plays a role in patient enrollment needs to be considered. Clinicians have used corticosteroids for the management of septic shock for more than 50 years. International surveys pre and post ADRENAL have demonstrated marked variability in practice in terms of the type, dose and the triggers for corticosteroid prescription in patients with septic shock [7, 8]. These findings suggest that there may be clinician biases in selection of patients for corticosteroid prescription in clinical practice and whether these decisions influenced enrolment into the ADRENAL trial is not known.

**Table 2. Multivariable logistic regression for eligible patients being declined for randomization by clinicians in the ADRENAL trial.**

| All sites model (n = 5329) | Odds Ratio (95% CI) | P* |
|---|---|---|
| Tertiary ICU (versus non-tertiary) | 0.89(0.35–2.31) | 0.8 |
| ANZ site (yes versus no) | 1.86(0.62–5.59) | 0.3 |
| Volume (per 1000 admissions/year) | 1.14(0.66–1.98) | 0.6 |
| First author site (yes versus no) | 0.29(0.03–3.13) | 0.3 |
| Management committee site (yes versus no) | 1.27(0.41–3.97) | 0.7 |
| Research coordinator FTE (per 1 FTE) | 0.99(0.73–1.34) | 1 |
| After-hours research coordinator availability (yes versus no) | 0.70(0.31–1.56) | 0.4 |
| ANZ sites only model (n = 4291) | Odds Ratio (95% CI) | p |
| Tertiary ICU (versus non-tertiary) | 0.52(0.17–1.59) | 0.3 |
| Volume (per 1000 admissions/year) | 1.15(0.52–2.55) | 0.7 |
| CTG membership (yes versus no) | 2.86(0.92–8.88) | 0.1 |
| First author site (yes versus no) | 0.27(0.03–2.61) | 0.3 |
| Management committee site (yes versus no) | 1.87(0.56–6.19) | 0.3 |
| Research coordinator FTE (per 1 FTE) | 0.99(0.63–1.55) | 1 |
| After-hours research coordinator availability (yes versus no) | 0.76(0.34–1.71) | 0.5 |
| Director of Research FTE (per 1 FTE) | 0.84(0.24–2.97) | 0.8 |

Definitions of abbreviations: ICU = intensive care unit; ADRENAL = Venkatesh et al. '*Adjunctive glucocorticoid therapy in patients with septic shock*' Trial; ANZ = Australia and New Zealand; FTE = full- time equivalent; CTG = Australia New Zealand Intensive Care Society Clinical Trials Group

* Each effect estimate listed in this table has been generated from one of two multivariable mixed effects model ("All sites model" and "ANZ sites only model") i.e., they are adjusted for all other variables listed in the table.

Whilst RCTs have traditionally been conducted in large tertiary centers [9], our results show that the participation of smaller non-tertiary centers not only add to patient enrollment, but have similar, comparable recruitment patterns.

We acknowledge that this study is limited by a lack of power, as suggested by the wide confidence intervals in the multivariable analysis. Hence, the findings presented here should be considered exploratory. Nonetheless, the ADRENAL trial was a pragmatic, multinational trial that is representative of modern, large-scale critical care trials and collected detailed screening logs from all recruiting sites. Combined with resourcing data from a high-quality binational registry, the findings presented are robust with respect to internal validity. To improve external validity, a larger sample of trials would be required to confirm our findings.

Further investigation into the factors influencing clinician decline rate is warranted. We recommend that screening logs collect additional data to include patient-related, intervention-related, consent-related, or clinician decision factors that lead to eligible patients not being randomized into clinical trials. The establishment of a 'baseline' clinician decline rate (due to non-modifiable factors) may be useful in benchmarking the performance of RCTs as well as individual trial sites. Given that some sites will be participating in multiple trials simultaneously, data on co-enrollment at a site-level would allow for comparisons of the clinician decline rate, not just between sites, but also between trials as well.

Monitoring the clinician decline rate over the course of a trial may inform trial management committees to address this issue on a real-time basis, so that interventions such as increased frequency and intensity of educational site visits can be effectively targeted.

## Conclusion

Clinicians declining to randomize patients into critical care RCTs has a substantive effect on patient enrollment, but this does not appear to be associated with individual site characteristics.

## Supporting information

**S1 File. List of participating sites.**
(PDF)

## Acknowledgments

The authors thank the College of Intensive Care Medicine of Australia and New Zealand, the Australia New Zealand Intensive Care Society Centre for Outcomes and Resource Evaluation management committee, and clinicians, data collectors, and researchers at the contributing sites listed in S1 File.

## Author Contributions

**Conceptualization:** Mahesh Ramanan, Laurent Billot, Balasubramanian Venkatesh.

**Data curation:** Dorrilyn Rajbhandari, Balasubramanian Venkatesh.

**Formal analysis:** Mahesh Ramanan, Laurent Billot.

**Investigation:** Mahesh Ramanan, Balasubramanian Venkatesh.

**Methodology:** Mahesh Ramanan, Laurent Billot, Dorrilyn Rajbhandari, John Myburgh, Balasubramanian Venkatesh.

**Project administration:** Mahesh Ramanan.

**Resources:** Mahesh Ramanan, Dorrilyn Rajbhandari, John Myburgh, Balasubramanian Venkatesh.

**Software:** Mahesh Ramanan.

**Supervision:** John Myburgh, Balasubramanian Venkatesh.

**Validation:** Mahesh Ramanan.

**Visualization:** Mahesh Ramanan.

**Writing – original draft:** Mahesh Ramanan.

**Writing – review & editing:** Mahesh Ramanan, Laurent Billot, Dorrilyn Rajbhandari, John Myburgh, Balasubramanian Venkatesh.

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
