## [Decision Letter · Decision Letter 0]

23 Apr 2021

PONE-D-21-04343

An evaluation of factors that may influence clinicians’ decisions not to enrol eligible patients into randomized trials in critical care.

PLOS ONE

Dear Dr. Ramanan,

Thank you for submitting your manuscript to PLOS ONE. After careful consideration, we feel that it has merit but does not fully meet PLOS ONE’s publication criteria as it currently stands. Therefore, we invite you to submit a revised version of the manuscript that addresses the points raised during the review process.

We commend the authors for their submission. Although the manuscript has merit, several items require clarification and / or revision if the manuscript is to meet criteria for publication in PLOS ONE:

- Address all queries posed by Reviewer 1

- Address all queries posed by Reviewer 2

- Address all items indicated by tracked changes in the attached Word document (PONE-D-21-04343-1 [suggested edits])

- Carefully proofread your final revised document for typos as well as errors of syntax and grammar

Please submit your revised manuscript within 45 days of the date of this letter.  If you will need more time than this to complete your revisions, please reply to this message or contact the journal office at plosone@plos.org. Please include the following items when submitting your revised manuscript:

We look forward to receiving your revised manuscript.

Kind regards,

Linda L. Maerz, MD

Academic Editor

PLOS ONE

Journal Requirements:

Thank you for providing the date(s) when patient medical information was initially recorded. Please also include the date(s) on which your research team accessed the databases/records to obtain the retrospective data used in your study.

Please provide a list of the names of the institutions where the data was collected from as a supplementary file.

We note that you have indicated that data from this study are available upon request. PLOS only allows data to be available upon request if there are legal or ethical restrictions on sharing data publicly. For information on unacceptable data access restrictions, please see http://journals.plos.org/plosone/s/data-availability#loc-unacceptable-data-access-restrictions.

4a) If there are ethical or legal restrictions on sharing a de-identified data set, please explain them in detail (e.g., data contain potentially identifying or sensitive patient information) and who has imposed them (e.g., an ethics committee). Please also provide contact information for a data access committee, ethics committee, or other institutional body to which data requests may be sent.

4b) If there are no restrictions, please upload the minimal anonymized data set necessary to replicate your study findings as either Supporting Information files or to a stable, public repository and provide us with the relevant URLs, DOIs, or accession numbers. Please see http://www.bmj.com/content/340/bmj.c181.long for guidelines on how to de-identify and prepare clinical data for publication. For a list of acceptable repositories, please see http://journals.plos.org/plosone/s/data-availability#loc-recommended-repositories.

Reviewers' comments:

Reviewer's Responses to Questions

**Comments to the Author**

1. Is the manuscript technically sound, and do the data support the conclusions?

Reviewer #1: Partly

Reviewer #2: Yes

2. Has the statistical analysis been performed appropriately and rigorously? 

Reviewer #1: Yes

Reviewer #2: Yes

3. Have the authors made all data underlying the findings in their manuscript fully available?

Reviewer #1: Yes

Reviewer #2: Yes

4. Is the manuscript presented in an intelligible fashion and written in standard English?

Reviewer #1: Yes

Reviewer #2: Yes

5. Review Comments to the Author

Reviewer #1: The authors' question is important and relevant.

It is logical to characterize the study sites as having trainees and research support to attempt to identify reasons why clinicians may not include eligible patients in a clinical trial.

However, the study is a bit superficial:

First - Only one trial (ADRENAL), though a large trial, was included in the analysis. Any trial specific factors cannot be isolated or assessed. In other words, it is difficult to say if the included research sites had specific protocol reasons to decline eligible patients. Including other trials would potentially provide additional clarity.

Second - Additional site-level characteristics are needed. It is a big jump to say that because there is no correlation between a site being academic or having research support that the clinicians are likely concerned about a lack of equipoise.

While it helps that the ADRENAL trial allowed deferred consent, the authors' conclusion is still based on speculation.

For example, in how many other trials of this size did the sites participate?

Were the rates of declined eligible patients similar for all studies?

This manuscript will be more robust with the answers to these questions.

Reviewer #2: The methods are described with adequate detail. The study is probably underpowered as indicated by the wide confidence intervals reported for many of the variables. Realizing that that the sample size was fixed by what was available in the RCT data base I did not see a description of how adequate power was estimated. Having an estimate of the sample size required to detect differences in outcome could inform future studies and place the current results in perspective.

6. PLOS authors have the option to publish the peer review history of their article (what does this mean?). If published, this will include your full peer review and any attached files.

Reviewer #1: No

Reviewer #2: No

---

## [Author Response · Author response to Decision Letter 0]

20 May 2021

Dear Prof Linda Maerz,

Thank you for your thoughtful review and for giving us an opportunity to submit a revised manuscript. We have addressed all the points raised by the Reviewers and Editors. Our responses are highlighted in yellow after each query. Tracked and clean versions of the manuscript are attached to the submission as requested.

We look forward to hearing the outcome of your review.

Yours sincerely,

Mahesh Ramanan on behalf of all authors

Date: 11 May 2021

Reviewer 1

It is logical to characterize the study sites as having trainees and research support to attempt to identify reasons why clinicians may not include eligible patients in a clinical trial.

However, the study is a bit superficial:

First - Only one trial (ADRENAL), though a large trial, was included in the analysis. Any trial specific factors cannot be isolated or assessed. In other words, it is difficult to say if the included research sites had specific protocol reasons to decline eligible patients. Including other trials would potentially provide additional clarity.

Thank you for your review and comments. It is true that trial-specific factors cannot be assessed with our study. Having access to a larger group of trials with similar data would be ideal. However, this was not the case for our study as we did not have access to such data. We have included a statement in the discussion to this effect. 

Second - Additional site-level characteristics are needed. It is a big jump to say that because there is no correlation between a site being academic or having research support that the clinicians are likely concerned about a lack of equipoise.

While it helps that the ADRENAL trial allowed deferred consent, the authors' conclusion is still based on speculation.

For example, in how many other trials of this size did the sites participate?

Were the rates of declined eligible patients similar for all studies?

That particular statement with regards to deferred consent and clinician equipoise is indeed speculative, as we have acknowledged in the text. We have added some further discussion to clarify this point.

The data concerning other trials (‘competing trials’) and particularly the rates of CDR in these trials was not available to us. The ADRENAL trial protocol did allow for co-enrollment with several other trials, so it is possible that there were sites which were recruiting in competing trials.

Reviewer 2

The methods are described with adequate detail. The study is probably underpowered as indicated by the wide confidence intervals reported for many of the variables. Realizing that that the sample size was fixed by what was available in the RCT data base I did not see a description of how adequate power was estimated. Having an estimate of the sample size required to detect differences in outcome could inform future studies and place the current results in perspective.

Thank you for your review and for raising the important issue of statistical power. We did not have a baseline clinician decline rate which could be used for power calculations, hence, a priori calculations were not performed. Our results may be used to inform sample size calculations for future studies. 

We have elected not to perform post hoc calculations of power as there is significant controversy in the biostatistics literature regarding the validity of such calculations. A recent publication by Zhang et al (Zhang Y, Hedo R, Rivera A, et al. Post hoc power analysis: is it an informative and meaningful analysis? General Psychiatry 2019;32:e100069. doi: 10.1136/gpsych-2019-100069) had the following concluding statements:

“Power analysis is an indispensable component of planning clinical research studies. However, when used to indicate power for outcomes already observed, it is not only conceptually flawed but also analytically misleading. Our simulation results show that such power analyses do not indicate true power for detecting statistical significance, since post hoc power estimates are generally variable in the range of practical interest and can be very different from the true power.

In this report, we focus on the relatively simple statistical model for comparing two population means of continuous outcomes. The same considerations and conclusions also apply to non-continuous outcomes and more complex models such as regression. In general, post hoc power analyses do not provide sensible results.”

Responses to Comments

Page 3

What does this mean?

This should be tertiary status which is defined in the Methods section.

Page 4

Requires a space between “We” and “defined”

This has been inserted.

What is the rationale for derivation of these ICU-level variables? Wouldn’t ICU staffing also be relevant? For instance—full-versus part-time intensivist staffing; physician versus advanced practice provider staffing; trainee presence; number of patients per intensivist, etc.

We selected these variables based on availability and relevance to Australia and New Zealand Intensive Care practice. All ICUs included in the trial in Australia and New Zealand are staffed by specialist Intensivists on-site during the day and on-call at night with on-site trainee presence all 24 hours. There are no advanced practice providers or nurse practitioners in ANZ ICUs. The seniority of trainee presence (i.e. basic trainee, advanced trainee, fellow) is determined by the College of Intensive Care Medicine Level as described earlier in the Methods section. The number pf patients looked after by intensivists in ANZ follow the recommended CICM guidelines as these sites are accredited by the CICM for the purpose of training. We chose volume of admissions as a marker of ICU size, and the other variables selected were resourcing variables that have a direct link to research i.e. availability of research coordinators, director of research etc.

Page 5

Percentage?

Yes. This has been changed.

Page 6

Please provide more detail pertaining to the rationale of this speculation.

Yes. We have now provided this detail in the preceding paragraph.

Need a space between “of” and equipoise.

This sentence has been removed.

Page 7

Need space between “but” and “have”

This has been inserted.

If the study is underpowered, please explain why the conclusions are valid.

Thanks. We have added further to the paragraph to explain that the findings are exploratory but also why they hold validity.

Please define this.

We have shortened the sentence for clarity.

Page 8

Need a space between “on” and “enrollment”

This has been inserted.

---

## [Editor Report · Decision Letter 1]

15 Jul 2021

An evaluation of factors that may influence clinicians’ decisions not to enrol eligible patients into randomized trials in critical care.

PONE-D-21-04343R1

Dear Dr. Ramanan,

We’re pleased to inform you that your manuscript has been judged scientifically suitable for publication and will be formally accepted for publication once it meets all outstanding technical requirements.

Kind regards,

Linda L. Maerz, MD

Academic Editor

PLOS ONE

Additional Editor Comments (optional):

The authors are to be commended for addressing the editorial requests in the revised manuscript.
---

## [Editor Report · Acceptance letter]

19 Jul 2021

PONE-D-21-04343R1 

An evaluation of factors that may influence clinicians’ decisions not to enroll eligible patients  into randomized trials in critical care. 

Dear Dr. Ramanan:

I'm pleased to inform you that your manuscript has been deemed suitable for publication in PLOS ONE. Congratulations! Your manuscript is now with our production department. 

Kind regards, 

on behalf of

Dr. Linda L. Maerz 

Academic Editor

PLOS ONE